# Berries from *Luzuriaga radicans* Ruiz & Pav.: A Southern Chile Climbing Shrub as a Source of Antioxidants Against Chronic Diseases

**DOI:** 10.3390/plants14162555

**Published:** 2025-08-17

**Authors:** Sebastian Scharf, Javier Romero-Parra, Peter Winterhalter, Alfredo Torres-Benítez, Recep Gök, Mario J. Simirgiotis

**Affiliations:** 1Institute of Food Chemistry, Technische Universitat Braunschweig, Schleinitzstrase 20, 38106 Braunschweig, Germany; sebastian.scharf@tu-braunschweig.de (S.S.); p.winterhalter@tu-braunschweig.de (P.W.); 2Departamento de Química Orgánica y Fisicoquímica, Facultad de Ciencias Químicas y Farmacéuticas, Universidad de Chile, Santiago 6640022, Chile; javier.romero@ciq.uchile.cl; 3Carrera de Química y Farmacia, Facultad de Ciencias, Universidad San Sebastián, General Lagos 1163, Valdivia 5090000, Chile; alfredo.torres@uss.cl; 4Instituto de Farmacia, Facultad de Ciencias, Universidad Austral de Chile, Valdivia 5110566, Chile

**Keywords:** endemic Chilean berries, *Luzuriaga radicans*, Alstroemeriaceae, antioxidant capacity, docking calculations, enzyme inhibition, carotenoid esters

## Abstract

In recent years, numerous studies have emerged on the biological activities of endemic berries from the Valdivian Forest and their potential for therapeutic use. However, some species appear to have been relatively neglected. The objective of this study was to conduct, for the first time, a phytochemical composition analysis of a hydroalcoholic extract of *Luzuriaga radicans* Ruiz & Pav. and to evaluate its potential as an antioxidant and as an enzyme inhibitor in relation to chronic non-communicable diseases. The berries were collected in the Saval Park in Valdivia and were subsequently extracted via sonication in ethanol/water. UHPLC-DAD, HPLC-APCI (+)-MS, and UHPLC-ESI (+)-TOF-MS analysis allowed for the identification of several carotenoid ester species. According to UHPLC-DAD, the sum of carotenoids yielded was 983.4 ± 26.3 mg/kg DW, while the concentration of the phenolic compounds was 9.33 ± 0.01 mg GAE/g dry fruit. The extract exhibited antioxidant properties by scavenging DPPH (2,2-diphenyl-1-picrylhydrazyl), ABTS (2,2-azinobis-(3-ethylbenzothioazolin-6-sulfonic acid)) radicals, and ferric reducing antioxidant power (FRAP). It also demonstrated cholinesterase enzyme inhibitor capacities (AChE and BuChE—IC_50_: 6.904 ± 0.42 and 18.38 ± 0.48 µg/mL, respectively). Docking calculations were additionally performed for a selection of compounds in the berries. The data obtained suggest that the hydroalcoholic extract of *L. radicans* possesses significant potential as a natural antioxidant and for the inhibition of enzymes, making it a promising candidate for the development of phytotherapeutic and nutraceutical products, especially as a supplement against chronic diseases.

## 1. Introduction

The Valdivian temperate rainforest is home to a variety of unique species, some of which produce edible berries with potentially beneficial compounds. However, these species are endangered by climate change, the growing population, and the timber industry [1,2]. Previous research has identified phenolic compounds and several anthocyanins in some endemic berries collected in the Valdivian Forest [3,4], in particular from *Azara serrata* Ruiz & Pav. Several rare, glycosylated anthocyanins were detected and quantified using UHPLC-DAD-TIMS-TOF-MS and demonstrated the inhibition of specific enzymes related to chronic non-communicable diseases (CNCDs) [5].

*Luzuriaga* is a small Gondwanean genus with three representative species in the Southern area of America (*L. Marginata* (Gaertner) Bentham, *L. Mradicans* R. & P., and *L. polyphylla* (Hooker) Macbride) and one species from Oceania (*L. parviflora* (Hooker) Kunth.) [6]. *Luzuriaga radicans* Ruiz & Pav. (with local names including quilineja, paupauhuén, or azahar) (Figure 1) is an evergreen semi-herbaceous vine. Its leaves are up to 4 cm long, with 4–13 parallel veins. It has fragrant flowers with 2–4 articulated peduncles. The plant is a climbing vine that grows in trees in the O’Higgins, Maule, Ñuble, Biobío, Araucanía, Los Ríos, Los Lagos, and Aysén regions of Chile. It grows in acidic Chilean soils that are rich in organic matter, particularly in shaded, moist environments, such as the rainforests of southern Chile, which comprise a maritime temperate climate with a significant amount of annual rainfall (around 3500 mm), as well as temperatures ranging from 5 to 13 °C. It has edible orange–red berries that can be used as antipyretics, while its stems have been used in crafting, particularly for basketry, brooms, and other household utensils, since the XVII century. However, it is now endangered as a result of overexploitation and habitat loss [1].

Thus far, there are no scientific studies exploring the chemistry and bioactivity of the edible fruits from *L. radicans*; therefore, it represents an intriguing opportunity to find new medicinal properties of the berries, as well as providing possibly significant chemotaxonomic contributions. One of the techniques employed in recent years for the metabolomic analysis of fruit extracts has been ultra-high-performance liquid chromatography (UHPLC) in combination with a diode array detector (DAD) and quadrupole time of flight mass spectrometry (Q-TOF-MS). This technology, which offers high resolution and sensitivity, has led to a major advance in the separation of bioactive compounds such as carotenoids in complex mixtures, as well as in the determination of their elemental composition and structural identification. Therefore, the integration of UHPLC with mass spectrometry has facilitated the continuous discovery and confirmation of the chemical fingerprint of various plants, thus validating their potential biological effect [7,8,9,10,11]. In addition, while ESI (HESI) thermal ionization and photoionization sources have improved sensitivity for apocarotenoids and xanthophyll esters, atmospheric pressure chemical ionization (APCI) remains the reference method, as the nonpolar polyene backbone is poorly ionized by ESI. Both methods were used in this study to analyze carotenoid-rich berries that had not been studied previously.

In this context, it is evident that forests represent a reservoir with an immense phytochemical richness and antioxidant potential (e.g., carotenoids, flavonoids, phenolic acids, and tannins) and have become relevant for studying the prevention and treatment of chronic diseases such as obesity, diabetes, cardiovascular pathologies, cancer, aging, and inflammation, among others [12,13,14,15]. Among the important therapeutic targets are cholinesterase enzymes, which are central regulators of cholinergic signaling with profound effects on neuronal, cardiovascular, metabolic, and immunological health. Their dysregulation underlies chronic diseases, from Alzheimer’s disease and dementia to metabolic syndrome, while selective inhibitors and/or modulators show therapeutic potential [16]. Antioxidants and cholinesterases are linked in a feedback loop where oxidative stress upregulates enzyme activity, and many antioxidants counteract both redox imbalance and cholinergic impairment mediated by this activity [17]. Carotenoids, which are fat-soluble pigments found in fruits, plants, algae, and certain microorganisms, have been shown to be moderate but promising inhibitors of AChE and BuChE. They attenuate cholinesterase activity and preserve synaptic acetylcholine levels, thus offering a complementary approach to traditional Alzheimer’s treatment [18].

This work aims to measure the proximate composition, metals, and enzyme inhibitors (glucosidase, amylase, BuChE, and AChE) and antioxidant properties of the orange berries, as well as to analyze, for the first time, the carotenoid profile using UHPLC-DAD, HPLC-APCI(+)-MS, and UHPLC-ESI(+)-TOF-MS.

## 2. Results and Discussion

*L. radicans* fruits were collected in Saval Park, Valdivia, Chile, in April 2023 (50 g). An ethanol–water 1:1 (*v*:*v*) extract was prepared via maceration and sonication using lyophilized berries. This extract was chemically characterized, and its functional properties were determined (Figure 2).

### 2.1. Metals and Proximate Composition

The results of the proximate composition (Table 1) showed that *L. radicans* fruits are rich in carbohydrates (65.5%) and have a low lipid (0.03%) and protein content (6.34%), while the mineral content showed that the fruits are rich in magnesium (6.82 mg/kg) and potassium (43.26 mg/kg) but lower in sodium. These properties make this berry very good for the elderly and better than other fruits, such as *Azara dentata* [4]. Mg and Ca are important minerals for supporting heart function and bone formation, metabolizing glucose, muscle relaxation, and memory. The results obtained regarding the proximate composition of these berries indicate that these fruits, which are generally undervalued and considered only for decorative purposes, could be used as an important source of minerals, fiber, and protein; these results may encourage their consumption.

The proximate composition found in *L. radicans* fruits is comparable to the nutritional profile of other important fruits and plant sources from different parts of the world, such as the fruits of *Vitex doniana* and *Saba comorensis* [19], *Annona muricata* [20], *Ammodaucus leucotrichus* [21], and *Vicia faba* [22], which reveal a similar macronutrient and carbohydrate content. Similar to *Lycium barbarum*, which has been extensively studied for its nutritional value [23,24], the fruits of *L. radicans* also exhibit a notable mineral profile. The high content of magnesium and potassium, combined with low levels of sodium, suggests a potential health benefit and establishes the *L. radicans* extract as a promising source of bioactive compounds for the development of functional foods and nutraceuticals.

### 2.2. Antioxidant Activity and Content of Phenolics and Carotenes

During oxidative stress, reactive oxygen species (ROS), such as hydroxyl radicals, and non-radical species, such as hydrogen peroxide, are produced [25]. These species can react with a wide range of molecules that are present in living cells, such as amino acids, sugars, lipids, proteins, and nucleic acids, leading to their oxidation and, consequently, pathological processes or alterations in foods or cosmetics [26]. Various methods are used to measure the antioxidant capacity of natural products and evaluate their potential use as antioxidants.

In this study, the total phenolic compounds according to spectrometry were discrete (9.33 ± 0.01 mg GAE/g dry weight), being lower than those of Valdivian *A. serrata* berries (57 mg GAE/g dry weight) [5], while the total carotenoids were deemed to be high, as measured using spectroscopy (79.0 ± 0.3 mg carotene/100 g fruit) and HPLC (98 mg/kg), compared to the orange cocona fruit CD1 ecotype (12.2 mg/kg fruits) [27]. The oxygen radical absorbance capacity (ORAC) is a fluorescent and sensitive method [28], while the ABTS assay is a reproducible technique that is used to evaluate the antioxidant properties of extracts by donating hydrogen atoms to form a non-radical molecule [29]. In this study, the *L. radicans* extract achieved an IC_50_ of 6.65 ± 0.5 μg/mL and 9.95 ± 0.05 μg/mL with the DPPH (2,2-diphenyl-1-picrylhydrazyl) and ABTS (2,2-azinobis-(3-ethylbenzothioazolin-6-sulfonic acid)) antiradical methods, respectively. As double bonds are added to a molecule, it can be better oxidized and can demonstrate an improved antioxidant activity; carotenoids have several double bonds and can be excellent antioxidants.

The antioxidant capacity of the extract can be compared with that of natural antioxidants that are used commercially (e.g., gallic acid, DPPH: IC_50_ = 4.32 ± 0.5; quercetin, IC_50_ = 12.23 ± 0.8 µg/mL). Furthermore, the ORAC and FRAP (108.9 ± 4.07 and 47.8 ± 0.01 μmol Trolox/g dry fruit, respectively) were lower than those of *A. serrata* (387 and 426 μmol Trolox/g dry fruit, respectively) [5] (Table 2); the ORAC is also higher than that of *Ugni molinae* berries (222 μmol TE/g dry fruit) [30]. As a result of these activities, the extract could be included in cosmetic, medicinal, or food products, with the aim of protecting them from oxidation, or for skin or body care to protect against the effects of free radicals.

### 2.3. Enzyme Inhibitory Properties

Several important enzymes related to CNCD were analyzed using the hydro-ethanolic extract of the endemic fruits of *L. radicans*. The results are shown in Table 2. Some of the enzymes implicated in the development of metabolic syndromes are glucosidases, amylases, and lipases. α-Glucosidase and α-amylase are important hydrolases that release glucose by digesting glycogen and starch; they are implicated in diabetes and various other diseases such as infections and cancer [31]. The inhibition of α-amylase and α-glucosidase slows carbohydrate digestion and absorption, subsequently suppressing postprandial hyperglycemia [32]. In this study, the *L. radicans* extract showed no inhibitory activity on these enzymes but showed activity against cholinesterases (AChE and BuChE). Cholinesterase enzymes play a fundamental role in the development of Alzheimer’s disease (AD), since they catalyze the hydrolysis and inactivation of the neurotransmitter acetylcholine, thus producing choline and acetate.

Cholinesterase inhibitors, such as some phenolic compounds, improve cholinergic function in AD, preserving acetylcholine levels; therefore, they are good for the symptomatic treatment of AD [33]. The inhibition of these enzymes could also be useful in cases of autism and schizophrenia [34], as well as in relation to dementia and Parkinson’s disease [35]. In this study, the most notable results were those of acetylcholinesterase (IC_50_: 6.904 ± 0.42), which was several times less active than the standard galantamine (IC_50_: 0.402 + 0.02), and butyrylcholinesterase (IC_50_: 18.38 ± 0.48); however, the inhibition of α-glucosidase and α-amylase was very low (IC_50_: more than 1000) compared to acarbose. Cholinesterase inhibition activity could lead to the discovery of a naturally occurring drug that could be useful for further treatment (Table 2).

For future studies, it is important to test new concentrations to validate the potential of the *L. radicans* extract in relation to the inhibition of the α-glucosidase and α-amylase enzymes; it is also important to isolate its major carotenoid compounds, which may show a greater inhibitory activity. On the other hand, the effect of the extract on the inhibition of cholinesterase enzymes, especially acetylcholinesterase, is highly promising, showing a possible focus for application in the regulation of cholinergic neurotransmission in diseases like Alzheimer’s. However, it is necessary to continue with the characterization and isolation of the active compounds of the extract and to fully elucidate their mechanisms of action. Similarly, it is also necessary to explore other biological properties of the *L. radicans* extract, e.g., anti-inflammatory properties, to corroborate other therapeutic effects of the natural extract.

### 2.4. Analysis of the Carotenoid Profile

#### 2.4.1. Chromatographic Analysis of Carotenoids

Carotenoids are a class of natural pigments that are known for their colors, ranging from orange to red and yellow, in relation to fruits, vegetables, and flowers, as well as for the provitamin A activity that some of them possess. The analysis of the carotenoid profile was performed using UHPLC-DAD, HPLC-APCI (+)-MSn, and UHPLC-ESI (+)-TOF-MS. Identification was performed for carotenoids and carotenoid esters using MSn and DAD data. Carotenoids have characteristic absorption spectra, so the wavelengths of the maxima were determined, and the %III/II-ratio was calculated. Molecular ions were determined using APCI(+)-MSn measurements. Thirty-nine peaks (Figure 3 and Table 3) were tentatively identified for the first time in the *L. radicans* extract using positive ionization mode. UHPLC-ESI(+)-TOF-MS enabled the high-resolution detection of 20 carotenoids. The absence of other compounds may be attributed to insufficient ionization under the applied ESI(+) conditions, which may limit the detection of less polar or poorly ionizing carotenoids. The carotenoids could also be identified according to their UV spectrum and molecular ions; data from the literature were used for qualifications. Compounds other than carotenoids from the edible hydroalcoholic extract, including some fatty acids, sugars, and phenolic compounds, were also analyzed with HPLC Q-TOF-MS (using ESI-MS negative mode); see Appendix A.

Several different carotenoids belonging to the carotenes, xanthophylls, and carotenoid ester groups were identified based on the DAD and MS data. A total of 14 compounds from the carotene group could be tentatively identified (peaks 23–28, ms4, 30–34, and 37). The largest group consists of (all-E)-β-carotene, γ-carotene, (all-E)-lycopene, and various cis-isomers of these carotenes (peaks 23, 25, 26, 28, 30–32, 34, and 37), which are characterized by an m/z of 537 and a characteristic absorption maximum. Peak 24 has an m/z of 539 and can be assigned to β-zeacarotene. Three isomers of ζ-carotene could be determined with an m/z of 541 (peaks ms4, 27, and 28) [36]. Carotenoids with the largest number in the fruit are free xanthophylls, with a total of 19 compounds (peaks 1, 3–5, 9–21, and 29). Peak 1 can be assigned to a coelution of (all-E)-lutein and (all-E)-zeaxanthin, while peak 10 can be assigned to (all-E)-β-cryptoxanthin. Peaks 3–5 and 9 have an m/z of 569; due to the absorption maxima for peaks 3 and 5 at λ = 466 nm, they could be cis isomers of lutein or zeaxanthin. A total of 13 peaks (4, 11–21, and 29) can be assigned to xanthophylls with an m/z of 553; the carotenoids α-cryptoxanthin, β-cryptoxanthin, 5,8-epoxy-α-carotene, 5,6-epoxy-β-carotene, and zeinoxanthin can be considered according to the literature. Meanwhile, according to our detected MS peaks (Table 3) that cannot be identified, xanthophyll (MW 552) can be assigned to α-cryptoxanthin, β-cryptoxanthin, 5,8-epoxy-α-carotene, 5,6-epoxy-β-carotene, or zeinoxanthin, while xanthophyll (MW 600) can be attributed to violaxanthin, auroxanthin, neoxanthin, or luteoxanthin.

The carotenoid profile of *L. radicans* revealed several esterified carotenoids in addition to the free xanthophylls and carotenes. The parent xanthophylls were tentatively identified as esters with (all-E)-violaxanthin (peaks 12, ms1, ms2, 31, and 33) [37,38,39]; (all-E)-β-cryptoxanthin (peaks 35 and 38) [40]; and (all-E)-antheraxanthin (peak ms3 and 32) [37,38] (Table 3). Further esters could be partially assigned based on the MS data, whereby peaks 6, 7, 8, 10, and 27 belong to xanthophylls with a MW of 600, which may be (all-E)-viola-xanthin, auroxanthin, neoxanthin, or luteoxanthin. Peaks 13, 33, and 37 can be assigned to esters of xanthophylls with a MW of 568, i.e., in the literature, zeaxanthin-dimyristate, zeaxanthin-dilaurate, and zeaxanthin myristate palmitate were tentatively identified [36,37]. The carotenoid esters identified in this study mainly contained the saturated fatty acids palmitic acid (C12:0) and myristic acid (C14:0), which were bound as mono- or diesters. In addition, three xanthophyll esters with capric acid (C:10) were detected with peaks 6, 7, and 27, while two xanthophyll esters containing palmitic acid (C:16) were detected with peaks 13 and ms2.

Carotenoids are lipid-soluble compounds that show antioxidant properties by scavenging radicals [41], e.g., the ABTS radical [42], especially due to the conjugated double bond system in the structure of carotenoids [43]; however, the main beta-carotene suffers from trans (E) to cis (Z) isomerization, while the (all-E)-form is the predominant isomer that is found in unprocessed carotene-rich plant foods [44]. The results of the radical scavenging assays (DPPH, ABTS, and ORAC), as well as the ferric reducing antioxidant power (FRAP) assay, support the functional relevance of carotenoids in contributing to the total antioxidant capacity of the berries. In particular, the ABTS and ORAC assays, which are more sensitive to lipophilic antioxidants, revealed relatively strong activity levels, suggesting a significant role of carotenoids in the radical quenching processes observed. Furthermore, the carotenoid profile of *L. radicans* berries was dominated by (all-E)-β-carotene and its cis-isomers, which were accompanied by several other carotenes, xanthophylls, and esterified forms. It is well established that different carotenoids vary in their antioxidant potential, depending on their structure and polarity. For example, certain (Z)-isomers of lycopene have been reported to exhibit a higher antioxidant activity than the (all-E) form, likely due to the enhanced accessibility of the double bond system [45]. Some articles also link higher carotenoid intakes and tissue concentrations with reduced cancer and cardiovascular disease risk [41]. However, some authors mention that dependence on chain length and the character of the terminal function varies the activity in the TEAC assay, with increasing activity as follows: β-apo-8′-carotenal < β-apo-8′-carotenoic acid ethyl ester < 6′-methyl-β-apo-6′-carotene-6′-one (citranaxanthin) [44].

#### 2.4.2. Qualitative Analysis of Carotenoids

The quantification of carotenoids from the *L. radicans* extract was performed using UHPLC-DAD, using β-carotene equivalents as an internal standard (Appendix A and Appendix A) [46]; Figure 4 shows the carotenoid concentrations in the pulp and skin of *L. radicans*. The carotenes have the highest quantitative proportion, with a content of 549.93 ± 22.95 mg/kg fresh weight and a proportion of 55.92% of the total content. The various xanthophylls have a total content of 340.42 ± 13.99 mg/kg fresh weight and a contribution of 34.62%. The xanthophyll esters represent the group with the lowest content, at 82.04 ± 2.21 mg/kg fresh weight and 8.34%. The carotenoids (15-Z)-β-carotene, (all-E)-β-carotene, γ-carotene, and (13-Z)-β-carotene, which have the highest content, all belong to the carotene group.

In Chile, studies related to species of the Alstroemeriaceae family are very scarce and have been limited to micropropagation and in vitro germination assays for the purpose of production and germplasm conservation [47,48,49]. However, the presence of carotenoids in *L. radicans* is similar to that of species in the genus *Lilium*, which contain β-carotene (1.22 ± 0.06 to 12.85 ± 0.31 μg/g) and (E/Z)-phytoene (1.92 ± 0.15–4.81 ± 0.80 μg/g) in variable concentrations [50], along with plants like *Zamia dressleri*, which have high carotenoid content at the beginning of development (0 to 178 days) expressed as 169 ± 6 to 105 ± 6 μg/g [51], *Sorbus aucuparia*, which reports 95.68 µg/g DM [52], *Moringa oleifera*, with contents ranging from 3.3 to 1.7 µg/g FW [53], and *Vaccinium floribundum*, with 5.94 µg/g DM of lutein, which stands out for its high antioxidant value [54]. In summary, a richness of carotenes and their derivatives exists in the chemical profiles of multiple plants from different genera and families, with diverse biological effects that justify their study.

### 2.5. Docking Simulations

Docking simulations were performed for the carotenoids shown in Figure 5. 9′-cis-β-carotene, 15′-cis-β-carotene, and β-zeacarotene were selected as representative samples of carotenoid molecules for the docking simulations, based on their structures. Therefore, we explored how simple structural modifications affect binding affinity and interaction with the target proteins acetylcholinesterase (TcAChE) and butyrylcholinesterase (hBChE).

The acetylcholinesterase catalytic site has been well characterized and features a narrow gorge of approximately 20 Å in length, which extends halfway into the protein before widening near the base. The catalytic activity is primarily mediated by a triad of amino acids—Ser200, Glu327, and His440—which function as a nucleophile and charge relay system. Additionally, an oxyanion hole formed by Gly 121, Gly 122, and Ala 204 stabilizes the tetrahedral intermediate, while the choline anionic subsites (Trp 86, Tyr 337, and Phe 338) act as a cation–π clamp for the quaternary ammonium acyl pocket. Furthermore, the peripheral anionic site (PAS)—comprising Trp286, Tyr72, Tyr124, and Asp74—is considered to be one of the initial docking sites; however, a nearby hydrophobic pocket, which is formed by 14 residues including Trp84, Tyr121, Trp279, Phe288, Phe290, Phe330, Phe331, and Tyr334, also plays a significant role in substrate binding and orientation. On the other hand, butyrylcholinesterase, like acetylcholinesterase, corresponds to a serine hydrolase that belongs to the esterase/lipase family and shares a similar structural classification [55,56,57]. Its catalytic triad consists of Ser198, Glu325, and His438. Although the catalytic site is hydrophobic, a key distinction between butyrylcholinesterase and acetylcholinesterase lies in the composition of the gorge-lining residues; in butyrylcholinesterase, several aromatic residues that are present in acetylcholinesterase are replaced by hydrophobic ones [57]. Based on this information, we selected the three hydrophobic carotenoids for docking simulations to derive relevant descriptors.

Each carotenoid was fully optimized; energetic minimizations and protonation or deprotonation were carried out using the LigPrep tool in Maestro Schrödinger suite v.11.8 (Schrödinger, LLC, New York, NY, USA) [58]. The selected carotenoid compounds, as well as the known cholinesterase inhibitor galantamine, were subjected to docking assays into the acetylcholinesterase and butyrylcholinesterase catalytic sites, aiming to analyze the molecular descriptors and obtain the energy docking descriptors. This step aimed to provide a rationale for the inhibitory activities observed with the previously mentioned carotenoids (Table 4).

#### 2.5.1. Acetylcholinesterase (TcAChE) Docking Results

The two selected carotenoids—9′-cis-β-carotene and 15′-cis-β-carotene—formed multiple hydrophobic interactions within the catalytic site of acetylcholinesterase, particularly with hydrophobic residues lining the surface of the catalytic gorge, which are key contributors to their inhibitory activity. Within acetylcholinesterase, both carotenoids adopt a comparable orientation, exhibiting a partially bent conformation because of the high rotational flexibility conferred by their hydrocarbon chains. The binding energies obtained for the two derivatives were notably favorable and comparable to the binding energy of the inhibitor (−12.989 kcal/mol; see Table 4). Owing to the nature of its chemical structure, the carotenoid 9′-cis-β-carotene displays multiple hydrophobic interactions with the enzyme’s amino acid residues, resulting in a binding energy of −9.780 kcal/mol. The hydrophobic interactions observed for this derivative, considering it lacks polar groups, involve the residues Tyr70, Trp84, Tyr121, Leu127, Val129, Tyr130, Phe331, Trp280, Tyr334, and Ile444 (Figure 6A). On the other hand, 15′-cis-β-carotene also exhibited hydrophobic interactions. The implicated amino acids were Trp84, Val236, Glu240, Trp279, Leu282, Pro283, Ile287, Phe290, Phe330, Phe331, and Tyr334 (Figure 6B). The binding energy was −11.356 kcal/mol, which corresponds to a better value compared to that of 9′-cis-β-carotene; this is likely due to the greater number of interactions formed by this second derivative.

#### 2.5.2. Butyrylcholinesterase (hBuChE) Docking Results

Butyrylcholinesterase belongs to the same structural class of proteins as acetylcholinesterase, both being members of the esterase/lipase family and classified as serine hydrolases [55,56]. Like acetylcholinesterase, this enzyme possesses a deep and narrow gorge lined with several hydrophobic residues [57]. Therefore, three carotenoids—9′-cis-β-carotene, 15′-cis-β-carotene, and β-zeacarotene—were subjected to docking assays with this enzyme. β-zeacarotene possesses a long aliphatic chain and only one cycloaliphatic substituent; thus, it was included to evaluate whether lipophilic hydrocarbons contribute to binding affinity, considering that it exhibits a cLogP value of 15.466. The binding energies from the docking assays of carotenoid compounds against butyrylcholinesterase exhibited a similar pattern to those observed with acetylcholinesterase. The three tested carotenoids showed binding energies comparable to that of galantamine (Table 4). Notably, 9′-cis-β-carotene stood out with an energy of −9.815 kcal/mol. Similarly to acetylcholinesterase, the carotenoids subjected to docking assays in butyrylcholinesterase predominantly engaged in hydrophobic interactions due to the nature of their structures, which lack polar groups. In this sense, 9′-cis-β-carotene performed several hydrophobic interactions with Ser198, Pro230, Trp231, Thr284, Leu286, Val288, Tyr396, Phe398, and Pro527 (Figure 6C).

On the other hand, 15′-cis-β-carotene and β-zeacarotene also exhibit distinct hydrophobic interactions with the active site residues of butyrylcholinesterase, demonstrating their potential for stable binding within the enzyme’s hydrophobic pocket. 15′-cis-β-carotene interacts hydrophobically with residues such as Trp82, Thr120, Leu125, Tyr128, Glu197, Trp231, Leu286, Val288, Phe298, and Phe329 (Figure 6D). These interactions indicate that this compound is likely positioned deep within the hydrophobic cleft of butyrylcholinesterase, engaging with both aromatic and aliphatic residues, stabilizing the molecule through Van der Waals interactions. Likewise, β-zeacarotene also forms hydrophobic contacts, albeit with a slightly different set of residues—Trp82, Trp230, Val280, Thr284, Leu286, Val288, Ala328, Tyr396, Tyr440, and Trp430 (Figure 6E). The presence of the Val and Leu residues supports a strong hydrophobic anchoring within the enzyme. Interestingly, the three compounds share interactions with Val288 and Leu286, which may represent key hydrophobic hot spots within butyrylcholinesterase. These shared residues could be central to the binding of hydrophobic molecules and may influence the affinity of similar compounds.

## 3. Materials and Methods

### 3.1. Chemicals, Reagents, and Materials

Ultrapure water, ethyl acetate, ethanol, Folin–Ciocalteau reagent, ascorbic acid, AlCl_3_, FeCl_3_, gallic acid, magnesium metal, CH_3_CO_2_K 1M, quercetin, dimethyl sulfoxide, FeSO_4_, 2,2-diphenyl-1-picrylhydrazyl (DPPH), 2,4,6, tripyridyl-s-triazine (TPTZ), 2,20-azo-bis (2-amidinopropane dichlorohydrate), and analytical-grade solvents were obtained from Merck^®^ (Darmstadt, Germany). Trolox, β-Apo-8′-carotenal, and DMSO with a purity higher than 95% were purchased from Sigma-Aldrich Chem. Co. (St Louis, MO, USA), Phytolab gmbH & Co. KG (Verstenbergsgreuth, Germany), or Extrasynthese (Genay, France). Acetylcholinesterase (TcAChE, EC 3.1.1.7), butyrylcholinesterase (hBuChE, EC 3.1.1.8), 4-nitrophenyldodecanoate, phosphate buffer, dinitrosalicylic acid, trichloroacetic acid (Merck, Darmstadt, Germany), fetal calf serum (FCS, Gibco, Grand Island, NY, USA), L-glutamine (Merck, Darmstadt, Germany), α-amylase, α-glucosidase, standard p-nitrophenyl-D-glucopyranoside, acarbose, sodium persulfate sodium carbonate, ferrous sulfate, sodium acetate, sodium sulfate anhydrous, and absolute ethanol (99.5%) were obtained from Sigma-Aldrich Chem. Co. (Sigma, St. Louis, MO, USA). Double-deionized water was obtained from Nanopure^®^ (Werner GmbH, Leverkusen, Germany). Methanol (HPLC grade) was purchased from Fisher Scientific (Loughborough, UK). Disodium hydrogen phosphate dihydrate (≥99.0%, p.a.) and citric acid (≥99.5%, p.a.) were obtained from Carl Roth GmbH & Co. KG (Karlsruhe, Germany). The solvents used for the UHPLC-DAD-TOF analyses were water (LC-MS grade) and methanol (UHPLC-MS-grade), which were purchased from TH. Geyer GmbH & Co. KG (Renningen, Germany).

### 3.2. Plant Material

Mature *L. radicans* (Figure 1) fruits were collected from Saval Park, Valdivia, Chile (-39.8068° S, 72.9162° W) in April 2023. The lyophilized (Labconco Freeze Dry Systems, Model 7670541 2.5 Liter Palo Alto, CA, USA, −50 °C, vacuum 0.13 barr) berries (50 g) were milled using a Grindomix blade mill (GM 200, Retsch, Haan, Germany) and stored in an ultrafreezer at −80 °C (Haier, biomedical model Dw86L388A, Qingdao, China).

### 3.3. Berry Extract Preparation

An ethanol–water 1:1 (*v*:*v*) extract was prepared via maceration and sonication (ultrasound probe SXSONIC Processor (Sonics, Inc., Shanghai, China)) at 25 kHz for 15 min using 1 g each of separated freeze-dried pulp and peel, with 10 mL ethanol (1 g, 10 mL, three times). To obtain a product after lyophilization (Labconco Freeze Dry Systems, Model 7670541 2.5 Liter Palo Alto, CA, USA, −50 °C, vacuum 0.13 barr), two gummy residues (0.12 g and 0.06 g, respectively) were used to perform the biological tests, as well as HPLC QTOF- MS in negative mode. To analyze the more specific analyses for carotenoids, an extraction was performed with dichloromethane (CH_2_Cl_2_)/ethyl acetate (EtOAc)/Methanol (MeOH) (50/25/25, *v*/*v*/*v*) [59]. For this purpose, 50 mg of freeze-dried and deseeded fruit was added to an Eppendorf tube containing 1 mL of DCM/EtOAc/MeOH (50/25/25, *v*/*v*/*v*) (0.1% BHT); then, 50 µL of ISTD β-Apo-8′-carotenal was added. The sample was vortexed for 1 min and then centrifuged for 5 min. A total of 500 µL was taken and put into an Eppendorf tube, along with 700 µL of McIlvaine buffer (pH = 7). The sample was mixed with the vortexer for 1 min and then centrifuged for 5 min. In total, 300 µL was removed, placed in a rolled-edge glass, and dried under N_2_.

For analysis, the extract was diluted in 0.3 mL tBME/MeOH (90/10, *v*/*v*) and filtered through a 0.2 µm PTFE syringe filter. The extraction was performed in triplicate. All steps for the carotenoid extraction were performed under red light (630 nm) to avoid carotenoid UV degradation.

### 3.4. Chemical Contents

#### 3.4.1. Determination of Proximate Composition

The proximate composition was determined using the freeze-dried parts (pulp and peels) of *L. radicans* according to the methods established by the Association of Official Agricultural Chemists (AOAC) [60] with some modifications [61]. The moisture content was analyzed by directly drying the sample in a circulating air oven. In total, 5 g of each freeze-dried sample was added to a pre-weighed porcelain crucible and maintained at 105 °C until a constant weight was reached. For ash analysis, 5 g of each sample was placed in a porcelain crucible before being carbonized and incinerated in a muffle furnace at 550 °C until only ash remained. For protein analysis, 0.4 g of each sample was weighed in filter paper, placed in Kjeldahl tubes, and treated with 10 mL of sulfuric acid and 2 g of a catalytic mixture (potassium sulfate and copper sulfate). Digestion was carried out in a digestion block at 450 °C until the solution became bluish green. After digestion, the samples were allowed to cool to ambient temperature before being transferred to clean Kjeldahl tubes and placed in a nitrogen distiller. In the distiller, 25 mL of 30% NaOH was added to each digested sample to initiate distillation. The distillate was collected in a 125 mL Erlenmeyer flask containing 10 mL of distilled water and 2 drops of phenolphthalein indicator, before being titrated with HCl. The lipid content was determined by directly extracting the sample with petroleum ether in a continuous Soxhlet extractor. For this purpose, 5 g of each sample was placed in Soxhlet extraction cartridges. The cartridge containing the sample was added to the extractor and allowed to reflux for approximately 8 h. After distillation, the petroleum ether was removed from the flask using a vacuum. The flask with the residue was dried in an oven at 105 °C for about 1 h before being cooled in a desiccator to room temperature and weighed. The levels of the minerals magnesium (Mg), sodium (Na), iron (Fe), calcium (Ca), zinc (Zn), potassium (K), copper (Cu), and manganese (Mn) were determined in mineral measurement apparatus (Varian AA240, Belrose, Australia) using atomic absorption spectroscopy, which was previously set with standard solutions with known amounts of the minerals being determined using flames of air–acetylene and nitrous oxide–acetylene; the latter was only used for calcium analysis. Hollow monometallic cathode lamps were used for each element analyzed. All analyses were performed in triplicate.

#### 3.4.2. Total Polyphenol and Carotene Quantification

Standardized protocols were used to determine the total phenolic content (TPC) and total carotenoid content (TCC). Absorbance was recorded in a multiplate reader (Synergy HTX, Billerica, MA, USA). The calibration curves were created using gallic acid and a beta-carotene standard [27]. The phenolic compound content and carotene content were expressed as μg of gallic acid equivalent (GAE) per mL (μg GAE/mL) and mg of beta-carotene per g of sample, respectively.

#### 3.4.3. Ultra High-Performance Liquid Chromatography (UHPLC) Diode Array Detector (DAD) Analysis for the Quantification of Carotenoids

For the quantitation of carotenoids, an Agilent 1290 Infinity II System (Agilent Technologies, Waldbronn, Germany) equipped with a binary solvent manager, an autosampler, a column heater, and a diode array detector was used. The column was an Accucore C30 column (150 × 3.0 mm, 2.6 µm), Thermo Scientific (Dreieich, Deutschland), with a column temperature of 14 °C, a column flow of 0.4 mL min^−1^, and an injection volume of 5 µL. The mobile phases consisted of methanol/water (87/13, *v*/*v*; eluent A) and methanol/tBME/water (90/7/3, *v*/*v*; eluent B), using the following gradient program: 0 min, 2% B; 2 min, 14.5% B; 5.5 min, 22% B; 37 min, 69% B; 38 min, 69% B; 39 min, 95% B; 42 min, 95% B; 43 min, 2% B; 48 min, 2% B. The UV-Vis spectra were obtained in the range of 200–600 nm, while the chromatograms were analyzed at λ = 470 nm. Quantification was performed using external calibration with (all-E)-β-carotene, which was recorded with 8 points from 1 to 500 mg/L, each measured 3-fold at λ = 470 nm (R2 = 0.9998). Carotenoids were quantified as (all-E)-β-carotene equivalents. Data analysis was performed using the OpenLab Chromatography Data System (CDS), ChemStation Edition, Version 3.4 (3.4.0) (Agilent Technologies, Waldbronn, Germany).

#### 3.4.4. HPLC-APCI(+)-MS^n^ Analysis for the Characterization of Carotenoids

For identification, the samples were analyzed on an Agilent 1100/1200 series (Waldbronn, Germany) HPLC system, consisting of a binary pump (G1312A), an autosampler (G1329A), a column oven (G1316A), and a diode array detector (G1315B), which was coupled with an ion-trap mass spectrometer (HCT Ultra ETD II, Bruker Daltonics, Bremen, Germany) using an APCI (atmospheric pressure chemical ionization) ionization source. The same column was used for quantitative analysis, using UHPLC-DAD with a column temperature of 22 °C. The mobile phases consisted of methanol/water (90/10, *v*/*v*; eluent A) and methanol/tBME (10/90, *v*/*v*; eluent B), using the following gradient program: 0 min, 2% B; 2.0 min, 14.5% B; 37 min, 69% B; 38 min, 69% B; 39 min, 95% B; 42 min, 95% B; 43 min, 2% B; 48 min; 2% B. The flow rate was 0.3 mL/min, and the injection volume was 5 µL. The diode array detector was operated in an acquisition range of 200–700 nm. The HPLC-MS runs were additionally monitored at λ = 470 nm. The APCI source was operated in positive mode, using nitrogen as a nebulizer (45 psi) and drying gas at a rate of 7.0 L/min (temp. 350 °C). The scan range was set between m/z 125 and 1250 using the ultra-scan mode with a mass scanning range of 26.000 m/z per second. MS1 parameters were as follows: voltage of high-voltage (HV) capillary −3500 V; HV end plate offset −500 V; trap drive 64.0; octopole Rf amplitude 0.0 Vpp; lens 1 −200.0 V; capillary exit −200.0 V; target mass 500; max. accumulation time 200.000 μs; ion charge control (ICC) target 100,000, which was taken as an average of four spectra; and the collision energy of 5 ev cone voltage was 20 V. The results were evaluated with the software Data Analysis 4.0 (Bruker Daltonics, Bremen, Germany).

#### 3.4.5. UHPLC-TOF-MS Analysis for the Characterization of Carotenoids

The chromatographic analysis was performed on an Agilent 1290 Infinity system, which included the same parts as the system used for quantitation. The column used was an Accucore C30 column (150 × 3.0 mm, 2.6 µm), Thermo Scientific (Dreieich, Deutschland). The mobile phases that were used were (A) methanol/water (90/10; *v*/*v*) and (B) tBME/MeOH/water (90/7/3; *v*/*v*/*v*), with a temperature of 14 °C, a flow of 0.4 mL min^−1^, and an injection volume of 5 µL. The gradient was the same as for the carotenoid analysis using UHPLC-DAD (Section 3.4.3).

For mass spectrometry, the system was a timsTOF equipped with an electrospray ionization source (Bruker Daltonik, Bremen, Germany). For ESI-positive measurements, the settings were as follows: a scan range of 100–1850 m/z; an inversed ion mobility range of 1/k0, 0.55–1.90 V* s/cm^−2^; a ramp time of 67.3 ms; a spectra rate of 13.64 Hz; a collision RF of 1100 Vpp; a transfer time of 65 µs; a capillary voltage of 4500 V; a nebulizing gas pressure of 2.20 bar (N2); a dry gas flow rate of 10 L min^−1^ (N2); a nebulizer temperature of 220 °C; and a collision energy of 10 eV. To calibrate the mass spectrometer and trapped ion mobility, the ESI-L Low Concentration Tuning Mix (Agilent Technologies, Waldbronn, Germany) was used. To operate the system, the Bruker Compass Hystar Version 6.2 and otofControl Version 6.2 (Bruker Daltonik, Bremen, Germany) software were used. For evaluating the analyses, Bruker Compass Data Analysis Version 5.3 (Bruker Daltonik, Bremen, Germany) was used.

### 3.5. Antioxidant Activity

The free radical scavenging and antioxidant capacity of the different extracts were determined with spectrophotometric methods using a microplate reader (Synergy HTX, Billerica, MA, USA).

#### 3.5.1. Oxygen Radical Absorbance Capacity (ORAC) Assay

The ORAC assay evaluates the radical scavenging capacity through the application of 2,2-Azo-bis (2-amidinopropane) dihydrochloride (AAPH) to the samples. The excitation and emission wavelengths were measured at λ = 485 and 530 nm, respectively, using Trolox for the calibration curve. The results are expressed in μmol Trolox/g of dry fruit [5]. A detailed protocol for this test can be found in the Appendix A.

#### 3.5.2. Ferric Reducing Antioxidant Power (FRAP) Assay

The FRAP assay was based on the reduction of the ferric 2,4,6-tripyridyl-s-triazine complex (Fe^3+^-TPTZ to Fe^2+^-TPTZ), which generates a blue coloration in the samples and was measured via spectrophotometry at λ = 593 nm using a Trolox standard curve. The results are expressed in μmol Trolox/g of dry fruit [5]. A detailed protocol for this test can be found in the Appendix A.

#### 3.5.3. DPPH Scavenging Activity

The 2,2-diphenyl-1-picrylhydrazyl (DPPH) radical was bleached, which turns colorless as antioxidants provide protons. The reaction was monitored via spectrophotometry at λ = 515 nm using a gallic acid standard curve. The results are expressed in μg/mL, denoting the half inhibitory concentration (IC_50_) [62]. A detailed protocol for this test can be found in the Appendix A.

#### 3.5.4. ABTS Scavenging Activity

The test was performed using ABTS 2,2-azinobis-(3-ethylbenzothioazolin-6-sulfonic acid) (Sigma Aldrich, St. Louis, MO, USA). The decrease in absorbance was recorded at 1 and 6 min after the start of the reaction, and the percentages of decolorization were subsequently calculated [29]. The assays were performed in triplicate. The IC_50_ value (which is the concentration that can eliminate 50% of the free radicals) was determined. A detailed protocol for this test can be found in the Appendix A.

### 3.6. Enzymatic Inhibitory Activity

#### 3.6.1. Acetylcholinesterase and Butyrylcholinesterase Inhibition Assays

The inhibitory activity of the cholinesterase enzymes was evaluated as described previously [5]. Briefly, a solution with 5-dithio-bis (2-nitrobenzoic acid) (DTNB) was prepared in Tris-HCl buffer (pH 8.0) containing 0.02 M MgCl_2_ and 0.1 M NaCl. Then, the hydroethanolic extract of Luzuriaga (50 µL, 2 μg/mL) was mixed in a 96-well microplate with 125 mL of DTNB solution, acetylcholinesterase (TcAChE), or butyrylcholinesterase (hBuChE) (25 mL). It was dissolved in Tris-HCl buffer (pH 8.0) and incubated for 15 min at 25 °C. The reaction was initiated by the addition of acetylthiocholine iodide (ATCI) or butyrylthiocholine chloride (BTCl) (25 µL). After 10 min of reaction, the absorbance at a wavelength of 405 nm was measured, and the IC_50_ (μg/mL) was calculated [5].

#### 3.6.2. α-Glucosidase Inhibition Assay

Solutions were read at λ = 415 nm in a microplate reader over a one-minute interval for a total of 20 min, employing an acarbose standard curve. The stock solution of the α-glucosidase enzyme was prepared in 2 mL at 20 U/mL of buffer for subsequent dilution. The results are expressed as (IC_50_) in μg/mL [28].

#### 3.6.3. α-Amylase Inhibition Assay

Solutions were read via spectrophotometry at λ = 515 nm using an acarbose standard curve. The α-amylase enzyme at a concentration of 0.5 mg/mL was placed in 5 mL of 20 mM phosphate-buffered solution at pH 6.9. The results are expressed in μg/mL (IC_50_) [28].

### 3.7. Docking Calculations

Docking calculations were performed for every carotenoid shown in Figure 6. Each molecule was fully optimized, and energetic minimizations and protonation or deprotonation were carried out using the LigPrep tool in Maestro Schrödinger suite v.11.8 (Schrödinger, LLC, New York, NY, USA) [58]. The partial charges, as well as the geometries of all compounds, were fully set using the DFT method, with B3LYP/6-311G+(d,p) being set as the standard basis [63,64] in Gaussian 09W software [65]. The crystallographic enzyme structures of *Torpedo californica* acetylcholinesterase (TcAChE; PDBID: 1DX6 code [66]), as well as human butyrylcholinesterase (hBChE; PDBID: 4BDS code [67]), were acquired from the Protein Data Bank RCSB PDB [68]. Enzyme optimizations were acquired using the Protein Preparation Wizard from Maestro software, where water molecules and ligands of the crystallographic protein active sites were removed. In the same way, all polar hydrogen atoms at pH 7.4 were added. Appropriate ionization states for acid and basic amino acid residues were explored. The OPLS3e force field was employed to minimize protein energy. The enclosing box size was set to a cube with sides of 26 Å in length. The presumed catalytic site of each enzyme in the centroid of selected residues was chosen, considering their accepted catalytic amino acids—Ser200 for acetylcholinesterase (TcAChE) [69,70], as well as Ser198 for butyrylcholinesterase (hBChE) [71,72]. The Glide Induced Fit Docking protocol was used for the final pairings [73]. Compounds were scored using the Glide scoring function in extra-precision mode (Glide XP; Schrödinger, LLC, New York, NY, USA) [74] and were picked according to the best scores and best RMS values (cutting criterion: less than 1 unit) to determine the potential intermolecular interactions between the enzymes and compounds, as well as the binding mode and docking descriptors. Complexes were visualized in the Visual Molecular Dynamics program (VMD) and Pymol [75].

### 3.8. Statistical Analysis

All assays were performed at least three times with three different samples. Each experimental value is expressed as mean ± standard deviation (SD). The statistical program InfoStat (student version, 2011) was used to assess the degree of statistical correlation between the different groups. Comparisons between groups were made using Tukey’s test (at *p* < 0.05).

## 4. Conclusions

In this study, the carotenoid composition of extracts from *L. radicans* was described for the first time, along with its potential use as an antioxidant and possible use against chronic non-communicable diseases. Overall, these findings expand our knowledge of secondary metabolites in native *Luzuriaga* species and validate their bioactivity, especially their potential as a supplement in aid of Alzheimer’s disease (inhibition of the AChE enzyme), laying the groundwork for future studies. Future studies should focus on evaluating the biological effects of the key pure unknown carotenoid compounds in animal or cellular models, along with pharmacodynamics and pharmacokinetic research to elucidate their mechanisms of action. Furthermore, the development of biotechnological and or chemical synthesis strategies could enhance their future applications.

## Figures and Tables

**Figure 1 plants-14-02555-f001:**
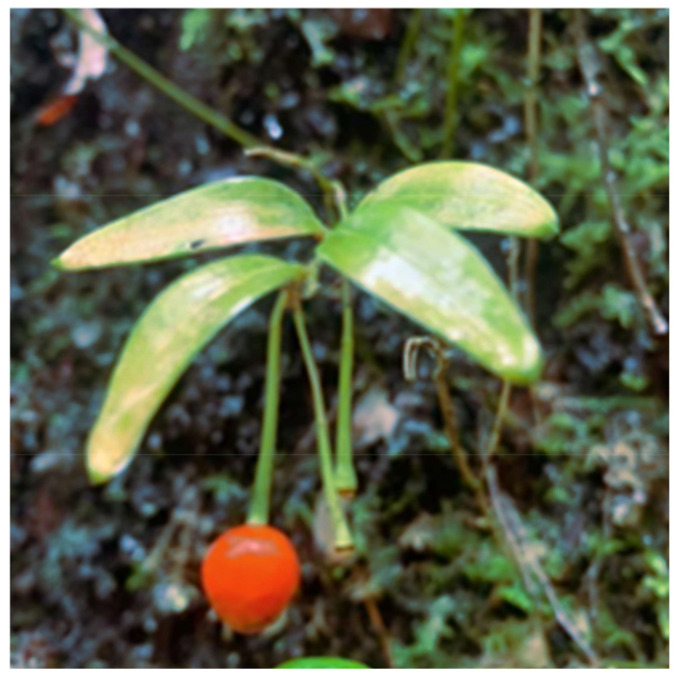
*Luzuriaga radicans* Ruiz & Pav. Saval Park, Valdivia, in April 2023.

**Figure 2 plants-14-02555-f002:**
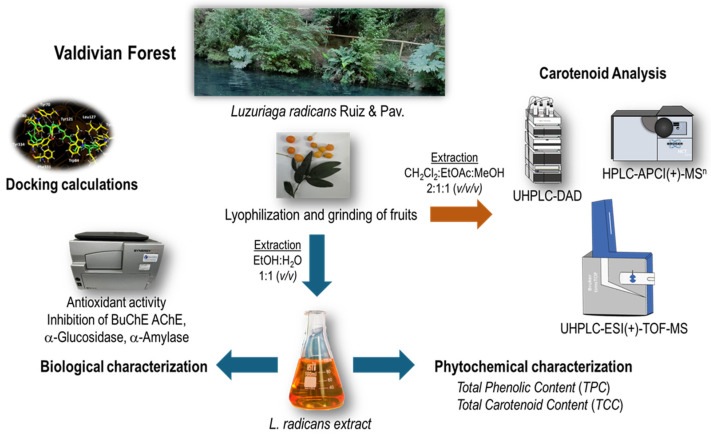
Flowchart for obtaining the *L. radicans* extract and its chemical and biological characterization.

**Figure 3 plants-14-02555-f003:**
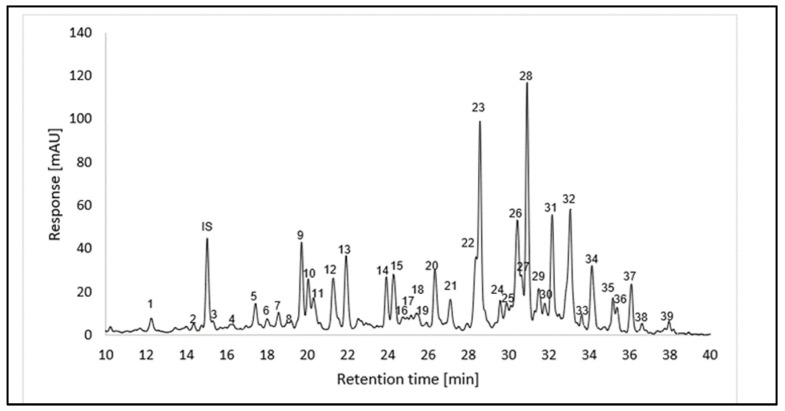
UHPLC-DAD chromatogram (recorded at λ = 470 nm) of the carotenoid extract from *L. radicans* (Accucore C30, 150 × 3.0 mm, 2.6 µm, Thermo Scientific (Dreieich, Deutschland)). For peak assignment, see Table 3.

**Figure 4 plants-14-02555-f004:**
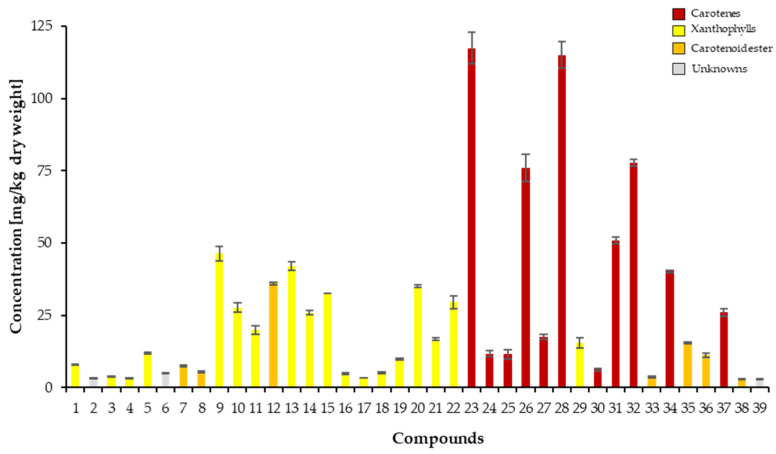
Carotenoid content in the pulp and skin of *L. radicans* berries, expressed as mg β-carotene equivalents per kg dry weight.

**Figure 5 plants-14-02555-f005:**
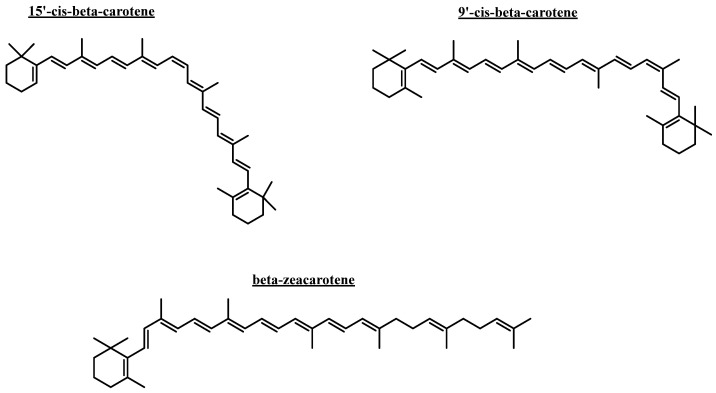
Compounds subjected to docking assays in the corresponding catalytic sites of acetylcholinesterase and butyrylcholinesterase.

**Figure 6 plants-14-02555-f006:**
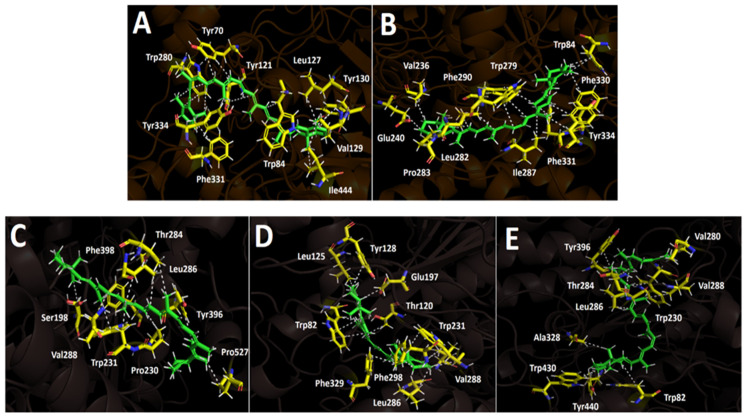
Predicted binding mode and predicted intermolecular interactions of different carotenoid compounds in the acetylcholinesterase catalytic site: (**A**) 9′-cis-β-carotene; (**B**) 15′-cis-β-carotene and butyrylcholinesterase catalytic site; (**C**) 9′-cis-β-carotene; (**D**) 15′-cis-β-carotene; (**E**) β-zeacarotene.

**Table 1 plants-14-02555-t001:** Proximate composition (in %) and mineral content (mg/kg) of *L. radicans*.

Proximate Composition		Mineral Content	
Humidity	12.2 ± 0.42 ^p^	Ca	3.35 ± 0.3 ^g^
Ashes	5.62 ± 0.16 ^o^	Mg	6.82 ± 0.2 ^h^
Total lipids	0.03 ± 0.001 ^c^	Fe	5.37 ± 0.3 ^i^
Crude protein	6.34 ± 0.2 ^d^	Zn	3.36± 0.1 ^g^
Crude fiber	10.45 ± 0.17 ^e^	Mn	1.22 ± 0.1 ^j^
Carbohydrates	65.5 ± 9.2 ^f^	Cu	0.67 ± 0.1 ^k^
		K	43.26 ± 0.3 ^l^
		Na	0.92 ± 0.1 ^m^

Each value represents the mean ± SD of three replicates (*n* = 3). Different letters within the same column indicate a significant difference according to the Tukey test at a 0.05 level of significance (*p* < 0.05).

**Table 2 plants-14-02555-t002:** Radical scavenging activity of 1,1-diphenyl-2-picrylhydrazyl radical (DPPH), ABTS radical, total phenolic content (TPC), total carotenoid content (TCC), cholinesterase inhibition capacity, and glucosidase and amylase inhibition capacity of the hydroalcoholic extract of *L. radicans*.

Sample	DPPH ^a^	ABTS ^a^	ORAC ^b^	FRAP ^b^	TPC ^c^	TCC ^d^	AChE ^e^	BuChE ^e^	α-Glucosidase ^e^	α-Amylase ^e^
Extract	6.65 ± 0.5	9.95 ± 0.05	108.9 ± 4.07	47.8 ± 0.01	9.33 ± 0.01	79.0 ± 0.3	6.904 ± 0.42	18.38 ± 0.48	>1000	>1000
Gallic acid	4.32 ± 0.5	16.7± 0.05	-	-	-	-	-	-	-	-
Acarbose	-	-	-	-	-	-	-	-	138.9 ± 0.01	10.04 ± 0.02
Galantamine	-	-	-	-	-	-	0.402 ± 0.02 ^e^	5.33 ± 0.01	-	-
Quercetin	12.23 ± 0.8	15.72 ± 0.05	-	-	-	-	-	-	-	-

^a^ DPPH antiradical and ABTS activities are expressed as IC_50_ in μg/mL. ^b^ Expressed as μmol Trolox/g dry fruit. ^c^ Total phenolic content (TPC) expressed as mg gallic acid equivalent GAE/g dry weight. ^d^ Total carotene content (TCC) expressed as mg beta-carotene 100 g dry weight. ^e^ Inhibitory enzymes of cholinesterases, α-glucosidase, and α-amylase enzymes in IC_50_ expressed in µg/mL. Values in the same column are significantly different (*p* < 0.05). Dash (-) means not tested (i.e., not applicable).

**Table 3 plants-14-02555-t003:** Carotenoid profiling data of *L. radicans*.

Peak ^a^	Compounds	R_t_ [min] ^b^	Molecular Formula	λ_max_ [nm] ^c^			III/II [%]	APCI(+)-MS^n^		ESI(+)-TOF-MS		
				I	II	III		m/z [M+H]^+^	MS^n^ Fragmentation m/z	Detected Mass m/z	Theor. Mass m/z	Mass Error [ppm]
1	(all-*E*)-lutein and (all-*E*)-zeaxanthin	12.24	C_40_H_56_O_2_	424	448	476	50	569.3	551.3 [M+H-18]^+^			
2	Unknown 1	14.34		-	450	-	-	591.3				
IS	*β*-Apo-8′-carotenal	15.01	C_30_H_40_O	464			-	417.2	399.1 [M+H-18]^+^, 325.1, 293.1	417.3150	417.3152	0.4
3	Ni ^d^, xanthophyll_MW ^e^ 568	15.30	C_40_H_56_O_2_	422	440	466	nd ^f^	569.3	551.3 [M+H-18]^+^; 429.2			
4	**ni, xanthophyll_MW 552 ^g^** ni, xanthophyll_MW 568	16.30	**C_40_H_56_O** C_40_H_56_O_2_	426	452	484	nd	**553.3** 569.3	535.3 [M+H-18]^+^; 495.2 551.3 [M+H-18]^+^; 429.2			
5	ni, xanthophyll_MW 568	17.42	C_40_H_56_O_2_	416	442	466	25	569.3	551.2 [M+H-18]^+^; 483.3			
6	ni, xanthophyll_MW 600 caprate	18.02	C_52_H_79_O_5_	398	424	446	83	755.4	737.4 [M+H-18]^+^; 645.4; 583.3 [M+H-172]^+^			
7	ni, xanthophyll_MW 600 caprate	18.57	C_52_H_79_O_5_	424	448	478	24	755.4	737.4 [M+H-18]^+^; 645.4; 583.3 [M+H-172]^+^			
8	ni, xanthophyll_MW 600 laurate	19.02	C_52_H_79_O_5_	418	442	468	20	783.5	765.5 [M+H-18]^+^; 583.3 [M+H-200]^+^; 565.3 [M+H-18-200]^+^			
9	ni, xanthophyll_MW 568	19.72	C_40_H_56_O_2_	296 436	460	490	58	569.3	551.3 [M+H-18]^+^; 483.2			
10	**(all-*E*)-*β*-cryptoxanthin** ni, xanthophyll_MW 600 laurate	20.04	**C_40_H_56_O** C_54_H_83_O_5_	422	444	474	42	**553.3** 783.5	535.3 [M+H-18]^+^ 765.4 [M+H-18]^+^; 583.2 [M+H-200]^+^; 565.3 [M+H-18-200]^+^			
11	ni, xanthophyll_MW 552	20.31	C_40_H_56_O	430	454	484	nd	553.3	535.3 [M+H-18]^+^	552.4318	552.4326	1.5
12	**(all-*E*)-violaxanthin laurate** ni, xanthophyll_MW 552	21.23	**C_52_H_79_O_5_** C_40_H_56_O_2_	425	448	478	5.0	**783.4** 553.3	765.5 [M+H-18]^+^; 673.3; 583.2 [M+H-200]^+^; 565.2 [M+H-18-200]^+^; 535.2 [M+H-18]^+^	553.4394	553.4404	1.8
13	**ni, xanthophyll_MW 552** xanthophyll_MW 568 myristate palmitate	21.92	**C_40_H_56_O** C_70_H_113_O_4_	420	440	468	nd	**553.3** 1017.6	535.3 [M+H-18]^+^ 999.6 [M+H-18]^+^	553.4393	553.4404	2.1
ms1	(all-*E*)-violaxanthin myristate		C_54_H_83_O_5_					811.7	793.7 [M+H-18]^+^; 583.2 [M+H-228]^+^; 565.3 [M+H-18-228]^+^			
14	ni, xanthophyll_MW 552	23.93	C_40_H_56_O	418	442	466	42	553.3	535.3 [M+H-18]^+^; 471.2; 429.2	553.4395	553.4404	1.6
15	ni, xanthophyll_MW 552	24.28	C_40_H_56_O	416	438	464	44	553.3	535.3 [M+H-18]^+^; 471.2; 429.2	553.4392	553.4404	2.2
16	ni, xanthophyll_MW 552	24.78	C_40_H_56_O	-	460	488	nd	553.3	535.3 [M+H-18]^+^; 493.2			
17	ni, xanthophyll_MW 552	24.96	C_40_H_56_O	436	454	486	nd	553.3	535.3 [M+H-18]^+^; 493.2			
18	ni, xanthophyll_MW 552	25.15	C_40_H_56_O	-	454	486	40	553.3	535.3 [M+H-18]^+^; 493.2			
19	ni, xanthophyll_MW 552	25.44	C_40_H_56_O	-	448	488	nd	553.3	535.3 [M+H-18]^+^; 493.2			
ms2	(all-*E*)-violaxanthin palmitate		C_56_H_87_O_5_					839.8				
20	ni, xanthophyll_MW 552	26.34	C_40_H_56_O	296 438	460	490	46	553.4	535.3 [M+H-18]^+^; 429.2; 385.2	553.4396	553.4404	1.4
21	ni, xanthophyll_MW 552	27.10	C_40_H_56_O	436	460	488	nd	553.4	535.3 [M+H-18]^+^; 461.3; 413.2			
ms3	(all-*E*)-antheraxanthin myristate		C_54_H_82_O_4_					795.7		795.6291	795.6286	−0.6
22	ni, xanthophyll_MW 552		C_40_H_56_O	362 440	468	498	55	553.3	535.3; 467.2 [M+H-18]^+^; 413.2	553.4399	553.4404	0.9
23	**(15-*Z*)-*β*-carotene** phytoene phytofluene isomer 1	28.56	**C_40_H_56_** C_40_H_64_ C_40_H_62_	418 274 332	440 286 348	468 298 367	nd	**537.4** 545.5 543.5	455.1; 413.2 463.2 [M+H-82]^+^ 461.5 [M+H-82]^+^; 337.3 [M-205]^+^	543.4902 536.4367	543.4924 536.4377	4.1 1.9
ms4	ζ-carotene isomer 1		C_40_H_60_					541.4	459.2 [M+H-82]^+^; 417.2			
24	***β*-zeacarotene** phytofluene isomer 2	29.56	**C_40_H_58_** C_40_H_62_	410 332	432 348	460 367	nd	**539.3** 543.5	457.2 5 [M+H-82]^+^; 389.2	538.4528	538.4533	0.9
25	ni, carotene_MW 536	29.89		416	438	460	nd	537.3	455.1; 413.2	536.4368	536.4377	1.6
26	13-cis-*β*-carotene	30.42	C_40_H_56_	338 422	444	472	34	537.3	444.3 [M+H-92]^+^; 347.2	536.4378	536.4377	−0.3
27	**ζ-carotene isomer 2** ni, xanthophyll_MW 600 caprate laurate	30.58	**C_40_H_60_** C_62_H_96_O_6_	380	402	426	nd	**541.4** 937.6	459.2 [M+H-82]^+^; 391.2 919.6 [M+H-18]^+^; 765.5; 737.4 [M+H-200]^+^; 547.3 [M+H-18-200]^+^	540.4684	540.4690	1.0
28	**(all-*E*)-*β*-carotene** ζ-carotene isomer 3	30.90	**C_40_H_56_** C_40_H_60_	428	452	476	14	**537.3** 541.3	444 [M+H-92]^+^; 413; 399; 347; 279	536.4382	536.4377	−1.0
29	ni, xanthophyll_MW 552	31.48	C_40_H_56_O	294 440	466	502	nd	553.3	535.3 [M+H-18]^+^; 413.2			
30	ni, carotene_MW 536	31.76	C_40_H_56_	430	458	484	30	537.3				
31	**(9-*Z*)-*β*-carotene** (all-*E*)-violaxanthin dilaurate	32.15	**C_40_H_56_** C_64_H_100_O_6_	422	448	472	88	**537.3** 965.7	455.2; 413.1 947.7 [M+H-18]^+^; 765.7 [M+H-200]^+^, 747.7 [M+H-18-200]^+^, 565.4 [M+H-200-200]^+^	536.4375	536.4377	0.2
32	***γ*-carotene** (all-*E*)-antheraxanthin-dilaurate	33.04	**C_40_H_56_** C_64_H_100_O_5_	436	462	492	47	**537.3** 949.7	455.2; 413.2 931.8 [M+H-18]^+^; 669.4; 599.3	536.4375	536.4377	0.3
33	**(all-*E*)-violaxanthin-laurate myristate** ni, xanthophyll_MW 568 dilaurate	33.60	**C_66_H_104_O_6_** C_64_H_100_O_4_	424	454	478	40	**993.7** 933.7	975.7 [M+H-18]^+^; 793 [M+H-200]^+^; 765 [M+H-18-200]^+^ 916.7 [M+H-18]^+^; 733.5 [M+H-200]^+^			
34	(*Z*)-Lycopene	34.12	C_40_H_56_	442	468	498	71	537.3	413.2	536.4377	536.4377	−0.2
35	(all-*E*)-*β*-cryptoxanthin laurate	35.16	C_52_H_78_O_2_	422	449	476	24	735.5	718.6 [M+H-18]^+^; 535.3 [M+H-200]^+^; 443.2 [M+H-92-200]^+^			
36	ni, xanthophyll_MW 552 laurate	35.37	C_52_H_78_O_2_	448	444	472	110	735.5	718.6 [M+H-18]^+^; 535.3 [M+H-200]^+^; 443.2 [M+H-92-200]^+^			
37	**(all-*E*)-lycopene** ni, xanthophyll_MW 568 dimyristate	36.09	**C_40_H_56_** C_68_H_108_O_4_	448	474	504	71	**537.4** 989.7	455.2; 413.2 933.7	536.4372	536.4377	0.8
38	(all-*E*)-*β*-cryptoxanthin palmitate	36.60	C_56_H_86_O_2_	424	452	478	88	791.6	535.3 [M+H-256]^+^			
39	Unknown 2	37.95		424	454	478	88					

^a^ Peak number as identified by carotenoid analysis using UHPLC-DAD (Figure 3). Compounds labeled as “ms” were detected exclusively using HPLC-APCI(+)-MS and could not be assigned to a specific DAD peak. ^b^ Retention time on Accucore C_30_-column (Figure 3). ^c^ Absorption maxima. ^d^ ni—not identified. ^e^ MW—molecular weight. ^f^ %III/II could not be calculated because of the poor resolution of the UV/Vis spectrum. ^g^ The compounds highlighted in bold represent the predominant ion species of the respective peaks.

**Table 4 plants-14-02555-t004:** Binding energies obtained from docking experiments of selected carotenoids, as well as the known inhibitor galantamine, over acetylcholinesterase (*Tc*AChE) and butyrylcholinesterase (*h*BChE).

Compound	Binding Energy (kcal/mol) Acetylcholinesterase	Binding Energy (kcal/mol) Butyrylcholinesterase
9′-cis-β-carotene	−9.780	−9.815
15′-cis-β-carotene	−11.356	−8.353
β-zeacarotene	-	−7.948
Galantamine	−12.989	−7.125

## Data Availability

The original contributions presented in this study are included in the article/Appendix A. Further inquiries can be directed to the corresponding authors.

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
