# Peer review of "Berries from Luzuriaga radicans Ruiz & Pav.: A Southern Chile Climbing Shrub as a Source of Antioxidants Against Chronic Diseases"

_plants, 2025, doi:10.3390/plants14162555_

Round 1
Reviewer 1 Report
Comments and Suggestions for Authors
The manuscript, titled "Berries from Luzuriaga radicans Ruiz & Pav.: A Southern Chilean Climbing Shrub as a Source of Antioxidants Against Chronic Diseases," fits the journal's scope. It concerns phytochemical studies of Luzuriaga radicans berries. The research is original and provides new insights into the chemical composition and biological activity of the analyzed extracts. The topic is interesting, but the manuscript requires revisions. I appreciate the work the authors put into their research, but some elements raise my doubts and require clarification and corrections. Detailed information is provided below:
It is unclear why the authors focused on the carotenoid group. It is necessary to explain why this group of metabolites was subjected to more detailed analysis (MS).
I ask the authors to explain why biological activity and total phenolic and carotenoid content were tested for the ethanol-water extract, while UHPLC analyses were performed for a completely different extract, significantly different in polarity. It is known that this makes it difficult to determine which compound composition is associated with biological activity. Why didn't the authors perform UHPLC analyses of the ethanol-water extract? Or why wasn't the activity of the less polar extract checked? In its current form, biological studies cannot be linked to chemical composition. I propose completing the missing analyses to provide a more comprehensive insight into the chemical composition and biological activity.
Another problem concerns the quantitative analysis of carotenoids. The determinations of individual compound contents were made using only a single standard substance, which provides absolutely no precise quantitative results, only estimates. A spectrophotometric assay is sufficient to estimate total carotenoid content. However, other standard substances are required for quantitative determinations. If the authors do not have other standards, please provide only the results for beta-carotene. Please limit the results for other compounds to qualitative determinations.
Other suggestions:
Please explain the abbreviations used as they appear in the text, starting with the abstract and continuing. For example, ORAC on line 98, DPPD on line 103, DCM on line 345, and so on.
If possible, please paste Figure 1 in better resolution because it is currently of poor quality.
There is unnecessary hyphenation on lines 22, 24, and 258.
Lines 27, 99 – “ABTS•+” - •+ should be superscripted.
Please standardize the units in the discussion of results, as the content is sometimes given in mg/100 g and elsewhere in mg/kg.
In some places, the authors use the abbreviation ml, and in others, mL – please standardize this.
Line 313 - "distilled and ultrapure water" obtained from Merck?
Line 137 - "Merck (Santiago de Chile)" - ?
Line 339 - "absolute ethanol" - what does this mean according to the authors? What was the specific ethanol concentration - 99%, 99.8%, or something else?
Line 527 - At what p-level were the tests performed?
Comments on the Quality of English LanguageThe article is generally understandable but requires linguistic correction due to grammar.
Author Response
Revisions are in blue
Reviewer 1
The manuscript, titled "Berries from Luzuriaga radicans Ruiz & Pav.: A Southern Chilean Climbing Shrub as a Source of Antioxidants Against Chronic Diseases," fits the journal's scope. It concerns phytochemical studies of Luzuriaga radicans berries. The research is original and provides new insights into the chemical composition and biological activity of the analyzed extracts. The topic is interesting, but the manuscript requires revisions. I appreciate the work the authors put into their research, but some elements raise my doubts and require clarification and corrections. Detailed information is provided below:
It is unclear why the authors focused on the carotenoid group. It is necessary to explain why this group of metabolites was subjected to more detailed analysis (MS).
I ask the authors to explain why biological activity and total phenolic and carotenoid content were tested for the ethanol-water extract, while UHPLC analyses were performed for a completely different extract, significantly different in polarity. It is known that this makes it difficult to determine which compound composition is associated with biological activity. Why didn't the authors perform UHPLC analyses of the ethanol-water extract? Or why wasn't the activity of the less polar extract checked? In its current form, biological studies cannot be linked to chemical composition. I propose completing the missing analyses to provide a more comprehensive insight into the chemical composition and biological activity.
Response: We agree with the kind reviewer, and thank you for the corrections. The main compounds are carotenoids; in fact the berries are orange due to the presence of the carotenoids. This is why we have performed specific carotenoid extraction to really quantify these compounds. This is the first time that this has been done for these berries. We have added another table containing the edible hydroalcoholic extract compounds detected in negative mode. The following phrase was added to the main article: Other compounds rather than carotenoids including some fatty acids, sugars and phenolic compounds were analyzed in HPLC Q-TOF-MS in ESI-MS negative mode in the edible hydroalcoholic extract, supplementary material figure S2 and Table S2)
Another problem concerns the quantitative analysis of carotenoids. The determinations of individual compound contents were made using only a single standard substance, which provides absolutely no precise quantitative results, only estimates. A spectrophotometric assay is sufficient to estimate total carotenoid content. However, other standard substances are required for quantitative determinations. If the authors do not have other standards, please provide only the results for beta-carotene. Please limit the results for other compounds to qualitative determinations.
We thank the reviewer and acknowledge the limitations of using a single standard for the quantification of structurally diverse carotenoids.
We added the following clarification to the manuscript at lines 151-160 and 221 “blue notes“...as authentic standard compounds were not available for all carotenoids.
Quantification in the study was performed using all-trans-β-carotene as the reference standard, since commercial standards are not available for all other carotenoids and carotenes represent the predominant subgroup among the identified carotenoids. The extinction coefficients for most carotenoids are similar due to their comparable conjugated double bond systems. For this reason, we consider that expressing carotenoid levels as β-carotene equivalents offers meaningful insight into the relative abundance of individual carotenoids in relation to the total carotenoid pool. For clarification, the caption of figure 4 was changed to “Carotenoid content in the pulp and skin of L. radicans berries, expressed as mg β-carotene equivalents per kg dry weight (dw).” The extinction coefficients for the all carotenoids are similar (extinction coefficient for the standard: The all-trans-β-carotene extinction coefficient, used in spectrophotometric analysis, is 2592 when measured at 450 nm in hexane, according to a study published by Taylor & Francis Online. This value is crucial for determining the concentration of β-carotene and other related carotenoids in solutions, so we have reported Carotenoids quantified as (all-E)-β-carotene equivalents. Other standards are similar and very difficult to obtain, while quantitative results are similar and well used all over. Detection: photodiode-array or UV–Vis set to 450 nm (scan 200–600 nm for identity).
Other suggestions:
Please explain the abbreviations used as they appear in the text, starting with the abstract and continuing. For example, ORAC on line 98, DPPD on line 103, DCM on line 345, and so on.
We agree with the reviewer. Thank you for the corrections. They have been made in the Abstract. DPPH, ABTS•+ radicals (those are names of radicals) and ferric reducing antioxidant power (FRAP) have been defined, along with the other abbreviations in the text (2,2-azinobis-(3-ethylbenzothioazolin-6-sulfonic acid) and so on).
If possible, please paste Figure 1 in better resolution because it is currently of poor quality.
We agree with the reviewer. We have made the recommended corrections.
There is unnecessary hyphenation on lines 22, 24, and 258.
We have corrected this.
Lines 27, 99 – “ABTS•+” - •+ should be superscripted.
We agree with the reviewer. We have made the recommended corrections.
Please standardize the units in the discussion of results, as the content is sometimes given in mg/100 g and elsewhere in mg/kg.
We have corrected this.
In some places, the authors use the abbreviation ml, and in others, mL – please standardize this.
We agree with the reviewer. Thank you for the corrections. They have been made.
Line 313 - "distilled and ultrapure water" obtained from Merck?
We have corrected this.
Line 137 - "Merck (Santiago de Chile)" - ? Merck has a brand in Santiago de Chile but was changed as Germany as suggested
We have corrected this.
Line 339 - "absolute ethanol" - what does this mean according to the authors? What was the specific ethanol concentration - 99%, 99.8%, or something else?
This has been corrected to 99.5 %
Line 527 - At what p-level were the tests performed?
(p < 0.05). We agree with the reviewer. We have made the recommended corrections.
Comments on the Quality of English Language
Reviewer 2
Explain abbreviations in abstract from line 28.
We have corrected this.
Introduction
Highlight in the introduction about usage and functional properties of processed products obtained from this berry.
There are no functional applications for this plant and its fruits. That is why we consider this work to be a contribution to basic research for future applications and studies.
Explain abbreviations in the line 59.
We agree with the reviewer. Thank you for the corrections. They have been made.
Materials and method
On what kind of soil this plant was grown? Insert it in manuscript.
We agree with the reviewer. The plant grows in temperate forests in Chile in acidic soils rich in organic matter. Thank you for the corrections.
What are climate conditions during vegetative period of plant which was studied in this manuscript? Insert in manuscript.
We agree with the reviewer. Thank you for the corrections. It grows best in shaded, moist environments, such as the rainforest of southern Chile.
What were conditions of liophilization process? Insert in manuscript.
We have made this change.
In the line 350 explain abbreviation.
We have made this change.
Collision energy and cone voltage are missing. Insert it.
We agree with the reviewer. Thank you for the corrections. We used a collision energy of 5 ev, and the cone voltage was 20 V.
What was standard in ABTS method? Insert it.
We agree with the reviewer. Thank you for the corrections. They have been made.
What was alcohol percent in the extract? Insert it in manuscript.
- radicans fruits was collected in Saval Park, Valdivia, Chile in april 2023. The lyophilized (Labconco Freeze Dry Systems, Model 7670541 2.5 Liter Palo Alto, CA, USA, -50 oC, vacuum 0.13 barr) berries (50 g) were milled using a Grindomix blade mill (GM 200, Retsch, Haan, Germany) and stored in ultrafreezer at -80 (Haier, biomedical model Dw86L388A, Qingdao, P.R. China). An ethanol:water 1:1 (v:v) extract was prepared by maceration and sonication (ultrasound probe SXSONIC Processor (Sonics, Inc, Shangai, China at 25 kHz for 15 min in the dark (1 g, 15 mL, three times) to obtain a gummy residue (0.12 g) after liophylization. This extract was chemically characterized, and its functional properties were determined (Figure 2).
Results and discussion
Explain abbreviation in the line 103.
We have made this change.
How did you obtain extract which results were present in the table 2? Highlight it in table.
The type of extract used for biological activities was added to the table.
Did you conduct quantification of carotenoids? Explain it in manuscript.
Yes, we performed a semi-quantification of the carotenoids was carried out with β-carotene as the standard compound. The results are expressed as β-carotene equivalents and are shown in Figure 4.
Highlight biological activity of carotenoids? Insert this in the discussion.
We have made this change.
Explain in the subsection 2.5.1. and 2.5.2. why did you choose for docking simulation these two carotenoids.
These were chosen by availability in the extract and to perform docking with enzymes.
This manuscript is missing advanced statistical analysis.
Triplicate measurements were performed to confirm analytical reproducibility. However, as the study focused on the detailed profiling of a single L. radicans berry sample set without experimental groups or treatment variations, the application of advanced statistical analysis was not appropriate for the study design.
Reviewer 3
Line 21: Did you mean "enzyme inhibitor"? Please be careful in the use of the English language within the text. Re-phrase the whole sentence accordingly.
We have made this change.
Introduction: This section is quite limited. It must illustrate more extensively and critically the limitations of the study. You must also include comments about cholinesterases and their connection to chronic diseases. Which is the effect of antioxidants on this matter? In addition, in Lines 55-56, do you mean scarce studies on bioactivity? As there are some of them, you should cite them as a reference.
Thanks to the reviewer for the corrections. There are no recent studies on bioactivities of this species so we have deleted “scarce”
Lines 70-80: Very limited comparison to literature data. You must evaluate the similarities and differences of your findings with various other literature studies and highlight with more criticality the potential and importance of these components.
The information was expanded and compared with other studies.
Lines 111-131: You must discuss in more detail all the information presented in Table 2.
The description of the results obtained in enzyme inhibition was expanded.
Table 2: You must explain in the Table's description the meaning of the dash "-" symbol. Why were many parameters not determined? Explain this in the text.
It means that this test does not apply to that biological parameter.
Table 3: Please explain the absence of some values in the column "ESI(+)-TOF-MS".
In addition to the HPLC-APCI(+)-MSn measurements, ESI(+)-TOF-MS was employed to obtain high-resolution mass spectral data for the carotenoids. However, due to the inherently low ionization efficiency of non-polar carotenoids under ESI positive-mode, the detectability of certain compounds was limited and as a result does not appear in the "ESI(+)-TOF-MS" column.
Lines 202-211: Please extend significantly the discussion on the connection of the detected carotenoids to the antioxidant activity based on your own findings. You have previously mentioned the antioxidant activities you've recorded. You must explain these tendencies based on the concentrations of carotenoids you've measured. Also, use literature to Explain in the subsection how the chemical structures of carotenoids contribute to antioxidant activity.
When double bonds are added to a molecule it can be better oxidized and can have more antioxidant activity. Carotenoids have several double bonds.
The manuscript contains a discussion of the connection of the detected carotenoids to the antioxidant activity including literature data. We expanded the discussion and added the following:
“The results of the radical scavenging assays (DPPH, ABTS, and ORAC), as well as the ferric reducing antioxidant power (FRAP) assay, support the functional relevance of carotenoids in contributing to the total antioxidant capacity of the berries. In particular, the ABTS and ORAC assays, which are more sensitive to lipophilic antioxidants, revealed relatively strong activity levels, suggesting a significant role of carotenoids in the radical quenching processes observed. Furthermore, the carotenoid profile of L. radicans berries was dominated by (all-E)-β-carotene and its cis-isomers, accompanied by several other carotenes, xanthophylls, and esterified forms. It is well established that different carotenoids vary in their antioxidant potential, depending on their structure and polarity. For example, certain (Z)-isomers of lycopene have been reported to exhibit higher antioxidant activity than the (all-E) form, likely due to enhanced accessibility of the double bond system (Bohm et al. 2002 - Trolox Equivalent Antioxidant Capacity of Different Geometrical Isomers of α-Carotene, β-Carotene, Lycopene, and Zeaxanthin).
Lines 223-230: You must give the actual values of the concentrations detected in the cited literature. Comments like "is comparable", "in significant concentrations" etc are very generalized. You should be specific.
We have included the following: L. radicans is similar to species of the genus Lilium, which contain β-carotene and (E/Z)-phytoene in high concentrations.
Section 2.5: Please also make critical comments on the importance of the enzyme catalytic sites and their structure on the catalysis of reactions. Which enzyme sites are the best to obtain better outcomes? Are there any research developments towards the development of engineered enzymes that have better catalytic activities?
We have added a paragraph regarding the chosen compounds in the text.
Line 342: What do you mean with the phrase "3 times in the dark)". Please revise the text to improve clarity.
This was deleted
Line 352: Red light? Please be more specific about what you mean and provide the wavelength (nm) of this light.
Corrected, under red light (630 nm) to avoid carotenoid uv degradation.
Lines 457-477: You must describe the main steps of each protocol used for the determination of antioxidant activity (ORAC, FRAP, DPPH, ABTS).
The detailed protocol was added in the supplementary materials.
Lines 489-497: You must describe the main steps of α-glucosidase and α-amylase activity protocols.
The detailed protocols were added in the supplementary materials.

Reviewer 2 Report
Comments and Suggestions for Authors
Dear Authors,
Please find my suggestions below.
Explain abbreviations in abstract from line 28.
Introduction
Highlight in the introduction about usage and functional properties of processed products obtained from this berry.
Explain abbreviations in the line 59.
Materials and method
On what kind of soil this plant was grown? Insert it in manuscript.
What are climate conditions during vegetative period of plant which was studied in this manuscript? Insert in manuscript.
What were conditions of liophilization process? Insert in manuscript.
In the line 350 explain abbreviation.
Collision energy and cone voltage are missing. Insert it.
What was standard in ABTS method? Insert it.
What was alcohol percent in the extract? Insert it in manuscript.
Results and discussion
Explain abbreviation in the line 103.
How did you obtain extract which results were present in the table 2? Highlight it in table.
Did you conduct quantification of carotenoids? Explain it in manuscript.
Highlight biological activity of carotenoids? Insert this in the discussion.
Explain in the subsection 2.5.1. and 2.5.2. why did you choose for docking simulation these two carotenoids.
This manuscript is missing advanced statistical analysis.
Author Response

(The authors gave the same response as above.)

Reviewer 3 Report
Comments and Suggestions for Authors
This was a well-presented study on the antioxidant potential of berries from Luzuriaga radicans Ruiz & Pav. The experimental research was extensive although the evaluation of the findings should have been more critical. Please see below my specific comments.
-Line 21: Did you mean "enzyme inhibitor"? Please be careful in the use of the English language within the text. Re-phrase the whole sentence accordingly.
-Introduction: This section is quite limited. It must illustrate more extensively and critically the limitations of the study. You must also include comments about cholinesterases and their connection to chronic diseases. Which is the effect of antioxidants on this matter? In addition, in Lines 55-56, do you mean scarce studies on bioactivity? As there are some of them, you should cite them as a reference.
-Lines 70-80: Very limited comparison to literature data. You must evaluate the similarities and differences of your findings with various other literature studies and highlight with more criticality the potential and importance of these components.
-Lines 111-131: You must discuss in more detail all the information presented in Table 2.
-Table 2: You must explain in the Table's description the meaning of the dash "-" symbol. Why were many parameters not determined? Explain this in the text.
-Table 3: Please explain the absence of some values in the column "ESI(+)-TOF-MS".
-Lines 202-211: Please extend significantly the discussion on the connection of the detected carotenoids to the antioxidant activity based on your own findings. You have previously mentioned the antioxidant activities you've recorded. You must explain these tendencies based on the concentrations of carotenoids you've measured. Also, use literature to explain how the chemical structures of carotenoids contribute to antioxidant activity.
-Lines 223-230: You must give the actual values of the concentrations detected in the cited literature. Comments like "is comparable", "in significant concentrations" etc are very generalized. You should be specific.
-Section 2.5: Please also make critical comments on the importance of the enzyme catalytic sites and their structure on the catalysis of reactions. Which enzyme sites are the best to obtain better outcomes? Are there any research developments towards the development of engineered enzymes that have better catalytic activities?
-Line 342: What do you mean with the phrase "3 times in the dark)". Please revise the text to improve clarity.
-Line 352: Red light? Please be more specific about what you mean and provide the wavelength (nm) of this light.
-Lines 457-477: You must describe the main steps of each protocol used for the determination of antioxidant activity (ORAC, FRAP, DPPH, ABTS).
-Lines 489-497: You must describe the main steps of α-glucosidase and α-amylase activity protocols.
Comments on the Quality of English LanguageProofreading of the manuscript must take place by an experienced researcher or a journal service. The use of the English language must be improved since there were grammatical and expression errors in various parts of the text.
Author Response

(The authors gave the same response as above.)

Round 2
Reviewer 1 Report
Comments and Suggestions for Authors The authors responded to all comments and made suggested corrections, provided clarifications, and supplemented the data. The article can be published in its current form.Author Response
thank you, we have added the reviewer 3 corrrections

Reviewer 2 Report
Comments and Suggestions for Authors
Dear Authors,
Thank you for revised version of manusript.
Author Response
thanks to reviewer 2, we have added corrections as reviewer 3 queries

Reviewer 3 Report
Comments and Suggestions for Authors
The manuscript has been significantly improved. However, you must re-consider the following prior to publication.
-Introduction: I'd suggest you increase the size of the Introduction, to highlight better the potential and advancements of UHPLC and Mass spectrometry, the methods you've chosen for the analysis in your study. Also, my query in the 1st round of Review regarding the addition of comments about cholinesterases and their connection to chronic diseases was not answered. Also, what is the effect of antioxidants on the latter?
-Line 65 of Revised: The phrase "scarce bioactivity" is still present in the Revised manuscript. Please correct the text in order to be clear whether there are other studies (in this case, you must cite them) or not at all (in this case, you must remove the word "scarce").
-Lines 288-294 of Revised: With my comment in the 1st Review Round, I meant that you must mention in the text the actual concentration values (i.e. numbers or range of numbers) in parentheses for each example you give. E.g. when you write that the concentrations of L. radicans are similar to Zamia dressleri, give also the values/numbers of the concentration levels in Zamia dressleri. This whole paragraph needs to be revised and written in more clarity, i.e. separate the info in more distinct sentences with detailed numerical information.
Author Response
Reviewer 3
Response is in blue
-Introduction: I'd suggest you increase the size of the Introduction, to highlight better the potential and advancements of UHPLC and Mass spectrometry, the methods you've chosen for the analysis in your study.
We add added a paragraph, thanks to the reviewer for the nice recommendations:
One of the techniques employed in recent years for the metabolomics analysis of fruit extracts has been Ultra-High-Performance Liquid Chromatography (UHPLC) in combination with Diode Array Detector (DAD) and quadrupole time of flight mass spectrometry (Q-TOF-MS). This technology, which offers high resolution and sensitivity, has led to a major advance in the separation of bioactive compounds such as carotenoids in complex mixtures, as well as in the determination of their elemental composition and structural identification. Therefore, the integration of UHPLC with mass spectrometry has facilitated the continuous discovery and confirmation of the chemical fingerprint of various plants, thus validating their potential biological effect [7,8,9,10,11]; In addition, while ESI (HESI) thermal ionization and photoionization sources have improved sensitivity for apocarotenoids and xanthophyll esters, atmospheric pressure chemical ionization (APCI) remains the reference method, as the non-polar polyene backbone is poorly ionized by ESI. Both methods were used in this study to analyze carotenoid-rich berries that had not been studied previously.
Also, my query in the 1st round of Review regarding the addition of comments about cholinesterases and their connection to chronic diseases was not answered. Also, what is the effect of antioxidants on the latter?
Thank you to the nice reviewer. We have added some paragraphs.
Among the important therapeutic targets are cholinesterase enzymes, which are central regulators of cholinergic signaling with profound effects on neuronal, cardiovascular, metabolic, and immunological health. Their dysregulation underlies chronic diseases, from Alzheimer's disease and dementia to metabolic syndrome, while selective inhibitors and/or modulators show therapeutic potential [16]. Antioxidants and cholinesterases are linked in a feedback loop where oxidative stress upregulates enzyme activity, and many antioxidants counteract both redox imbalance and cholinergic impairment mediated by this activity [17]. Carotenoids, which are fat-soluble pigments found in fruits, plants, algae, and certain microorganisms, have been shown to be moderate but promising inhibitors of AChE and BuChE. They attenuate cholinesterase activity and preserve synaptic acetylcholine levels, thus offering a complementary approach to traditional Alzheimer's treatment [18].
-Line 65 of Revised: The phrase "scarce bioactivity" is still present in the Revised manuscript. Please correct the text in order to be clear whether there are other studies (in this case, you must cite them) or not at all (in this case, you must remove the word "scarce").
Deleted scarce sorry. (No papers can be found related to bioactivity nor ancient books).
-Lines 288-294 of Revised: With my comment in the 1st Review Round, I meant that you must mention in the text the actual concentration values (i.e. numbers or range of numbers) in parentheses for each example you give. E.g. when you write that the concentrations of L. radicans are similar to Zamia dressleri, give also the values/numbers of the concentration levels in Zamia dressleri. This whole paragraph needs to be revised and written in more clarity, i.e. separate the info in more distinct sentences with detailed numerical information.
Thanks to the reviewer for the nice recommendations. We have added the values requested.
In Chile, studies related to species of the Alstroemeriaceae family are very scarce and have been limited to micropropagation and in vitro germination assays for the purpose of production and germplasm conservation [47,48,49]. However, the presence of carotenoids in L. radicans is similar to that of species in the genus Lilium, which contain β-carotene (1.22 ± 0.06 to 12.85 ± 0.31 μg/g) and (E/Z)-phytoene (1.92 ± 0.15 – 4.81 ± 0.80 μg/g) in variable concentrations [50], along with plants like Zamia dressleri, which have high carotenoid content at the beginning of development (0 to 178 days) ex-pressed as 169 ± 6 to 105 ± 6 μg/g [51], Sorbus aucuparia, which reports 95.68 µg/g DM [52], Moringa oleifera, with contents ranging from 3.3 to 1.7 µg/g FW [53], and Vaccinium floribundum, with 5.94 µg/g DM of lutein, which stands out for its high antioxidant value [54]. In summary, a richness of carotenes and their derivatives exists in the chemical profiles of multiple plants from different genera and families, with diverse biological effects that justify their study.
